# Inferring Class Label Distribution of Training Data from Classifiers: An Accuracy-Augmented Meta-Classifier Attack

**Raksha Ramakrishna**     **György Dán**
Division of Network and Systems Engineering, EECS,
KTH Royal Institute of Technology,
Stockholm, Sweden
`rakshar,gyuri@kth.se`

## Abstract

Property inference attacks against machine learning (ML) models aim to infer properties of the training data that are unrelated to the primary task of the model, and have so far been formulated as binary decision problems, i.e., whether or not the training data have a certain property. However, in industrial and healthcare applications, the proportion of labels in the training data is quite often also considered sensitive information. In this paper we introduce a new type of property inference attack that unlike binary decision problems in literature, aim at inferring the class label distribution of the training data from parameters of ML classifier models. We propose a method based on *shadow training* and a *meta-classifier* trained on the parameters of the shadow classifiers augmented with the accuracy of the classifiers on auxiliary data. We evaluate the proposed approach for ML classifiers with fully connected neural network architectures. We find that the proposed *meta-classifier* attack provides a maximum relative improvement of $52\%$ over state of the art.

## 1 Introduction

Classification using machine learning (ML) models is increasingly considered for use in a variety of industrial applications, including outlier detection in manufacturing, bank fraud detection, and various diagnostics applications in healthcare. High quality training data plays a pivotal role in producing good models that can provide accurate classification results, but producing high quality labeled data sets is often expensive, making trained models a valuable asset.

At the same time, it is widely understood that ML models, particularly deep neural networks (DNNs), may learn information about the data they were trained on, and could be used by an adversary for identifying properties of the training data set by analyzing the parameters of the trained model. Such attacks, termed property inference attacks Melis et al. [2019], have been widely studied from the perspective of the privacy and confidentiality of the input features Rigaki and Garcia [2020]. There has been significantly less focus on property inference attacks against training labels, i.e., their distribution in the training data set. Notwithstanding, the class label distribution could be highly sensitive in industrial settings as it could reveal to competitors, e.g., the credibility of loans of banking customers or the production quality of assembly lines, with significant impact on a company's valuation.

In this work, we explore such property inference attacks on ML models trained for classification. The goal of the adversary is to infer the class label distribution of the training data from the trained ML classifier model. We consider white-box attacks, i.e., the adversary knows the model architecture and can access the parameters of the target classifier model.

2022 Trustworthy and Socially Responsible Machine Learning (TSRML 2022) co-located with NeurIPS 2022.

To motivate further this new class of property inference attack, we discuss two real-world examples where such attacks could cause havoc. Consider for example the manufacturing context, where a power tool vendor could train a classifier for identifying faulty bolt tightening using data from its customers (bolt tightening is often safety critical, e.g., in the car industry) for accelerating production quality monitoring. The proposed type of attack would make the trained model reveal information about the production quality of the customer whose data were used for training, which can be used by competitors for blackmailing and negative publicity. In lack of a countermeasure, the feasibility of the attack is considered a showstopper for this use case of ML. Another example is a bank that trains a classifier for predicting if a loan would be in default. Our proposed attack could be used for inferring the fraction of default loans of the bank based on the trained classifier, which could be used for affecting the bank's public standing. Importantly, these use cases involve binary classification, where the proposed attack performs best.

In this paper, we firstly define this new property inference attack and advocate the use of a shadow-training methodology Ateniese et al. [2015] for class label distribution inference. Shadow-training can be interpreted as a supervised-learning approach to property inference by training a 'meta-classifier' that takes model parameters as input and outputs the inferred property. In order to carry out shadow-training, a dataset of trained classifiers and the corresponding property is used. We propose a meta-classifier whose architecture is invariant to permutations of the neuron connection weights. Furthermore, the meta-classifier also takes as input the accuracy of the trained classifier on auxiliary data. We investigate the performance of the meta-classifier and its dependence on the selection of the shadow-training dataset. We find that the proposed meta-classifier can successfully infer the class label distribution and that accuracy augmentation is beneficial. We show empirical results for 2 datasets in the case of binary classifiers. The empirical results for multi-class classifiers is discussed in Appendix A.1. In Appendix A.2, we find that the attack is robust to random oversampling of the minority class, a technique to combat class imbalance.

## 1.1   Related work

Property inference attacks have traditionally been posed as binary classification problems in the literature  Melis et al. [2019], and are often based on meta-classifier models irrespective of the target classifier model architecture Parisot et al. [2021], Ganju et al. [2018], Ateniese et al. [2015], Suri and Evans [2021], Zhou et al. [2021]. The meta-classifier is trained using *shadow-training*, i.e., the adversary trains *shadow classifiers* using labeled shadow-training data sets obtained from auxiliary data with and without a certain property. The shadow classifiers have the same architecture as the target classifier. The input to the meta-classifier is a set of features from the trained shadow-classifier models, which are functions of the architecture and the parameters of the shadow classifier models. The meta classifier is then trained for binary classification of the chosen property. Recent work proposed to poison the training data set so as to improve attack performance Chase et al. [2021], but the resulting property inference attack is limited to distinguishing between two possible outcomes as is the case with rest of the literature in property inference attacks. Contrary to previous work on property inference attack, our focus is on inferring the class label distribution, which can be seen as a regression problem faced by the adversary, and hence distinguishes our work from binary classification-based property inference attacks prevalent in the literature. Even for a binary classification problem, class-label distribution inference is more difficult than property inference since the adversary wants to infer more fine-grained information about the dataset than the presence or absence of a property.

Closely related to our work is Zhang et al. [2021], where authors proposed a meta-classifier attack for inferring the distribution of sensitive features, using an ensemble of binary classifiers trained to decide whether the attribute appears in $10, 30, 50, 70, 90\%$ of the dataset, but this approach does not yield an accurate estimate of the label distribution. In the context of online learning, yet closely related to our work is Salem et al. [2020], where authors study the reconstruction of samples and dataset inference, and solve a class label distribution inference problem similar to ours. They do, however, assume that the attacker has black-box access to the model before and after model updates, which makes their attack model differ significantly from ours.

## 1.2 Organization

The rest of the paper is organized as follows. In Section 2 we describe the problem formulation, and in Section 3 we describe the proposed accuracy-augmented meta-classifier attack . In Section 4 we present numerical results for the UCI Census income and the MNIST data sets. Section 5 concludes the paper and discusses future work.

## 2 Problem formulation

Consider a target classifier whose task given an input $\boldsymbol{x} \in \mathbb{R}^D$ is to identify the most suitable class for the input amongst $C$ classes. Let the output of the classifier be a vector of length $C$ denoted by $\boldsymbol{y}$, where each entry $[\boldsymbol{y}]_i$ is the probability that the input belongs to a particular class $c$, $c = 1, 2, \ldots C$, i.e., $\sum_i [\boldsymbol{y}]_i = 1$. Therefore, the classifier output $\boldsymbol{y}$ lies in the probability simplex $\boldsymbol{\Delta}^{C-1}$, i.e., $\boldsymbol{y} \in \boldsymbol{\Delta}^{C-1}$, also called the the $C - 1$ unit simplex. We model the target classifier as a function $\boldsymbol{f}_t : \mathbb{R}^D \to \boldsymbol{\Delta}^{C-1}$ parametrized by vector $\boldsymbol{\theta}$. Then, the output of the classifier is $\boldsymbol{y} = \boldsymbol{f}_t(\boldsymbol{x}; \boldsymbol{\theta})$. The classifier $\boldsymbol{f}_t$ described above is trained on a labeled dataset, $\mathcal{D} \triangleq \{\boldsymbol{x}_i, \boldsymbol{e}_i\}_{i=1,2,\ldots N}$ consisting of $N$ data points where class label $\boldsymbol{e}_i \in [0, 1]^C$ is a coordinate vector so that the entry corresponding to class $c$ is 1 and the rest are zero. We denote by $N_c$ the number of training samples with class label $c$. Then, the class label distribution of the training data is defined as

$$\boldsymbol{p}_t = \frac{1}{N} \sum_{i=1}^N \boldsymbol{e}_i \in \boldsymbol{\Delta}^{C-1}, i.e., \quad [\boldsymbol{p}_t]_c \triangleq \frac{N_c}{\sum_{c=1}^C N_c}. \tag{1}$$

We consider cross entropy to be the loss function of the target classifier as it is commonly used in classification problems,

$$\mathcal{L}(\boldsymbol{\theta}) = -\sum_{i=1}^N \sum_{c=1}^C [\boldsymbol{y}_i]_c \log \left( [\boldsymbol{f}_t(\boldsymbol{x}_i; \boldsymbol{\theta})]_c \right). \tag{2}$$

The set of weights and biases $\boldsymbol{\theta}$ are updated by minimizing the loss function in (2).

### 2.1 Architecture of the target classifier

In this work, we consider classifiers that are fully connected neural networks. Let $L$ denote the number of layers after the input, by $m_\ell$ the neurons in layer $\ell$, and by $\mathbf{W}_1 \in \mathbb{R}^{D \times m_1}, \mathbf{W}_2 \in \mathbb{R}^{m_1 \times m_2}, \ldots \mathbf{W}_\ell \in \mathbb{R}^{m_{\ell-1} \times m_\ell}, \ldots, \mathbf{W}_L \in \mathbb{R}^{m_L \times C}$ the connection weight matrices betweeen adjacent layers. Let the vector $\boldsymbol{b}_\ell \in \mathbb{R}^{m_\ell}$ denote the biases of neurons in layer $\ell$, and by $\sigma_\ell(.)$ the activation function for neurons in layer $\ell$. The activation function for the last layer $L$ is sigmoid (or softmax for multi-class) so that the output $\boldsymbol{y}$ of the classifier is normalized to be a probability mass function. Thus, the parameters of the classifier function $\boldsymbol{f}_t$ consist of the elements of connection weight matrices and the biases. We can then represent parameter $\boldsymbol{\theta} \in \mathbb{R}^M$ by concatenating all the vectorized weight matrices, $\text{vec}(\mathbf{W}) = [\mathbf{w}_{11} \quad \mathbf{w}_{21} \quad \ldots \quad \mathbf{w}_{mn}], \mathbf{W} \in \mathbb{R}^{m \times n}$, and biases, i.e.,

$$\boldsymbol{\theta} \triangleq \left[ \text{vec}(\mathbf{W}_1) \ \text{vec}(\mathbf{W}_2) \ \ldots \ \text{vec}(\mathbf{W}_L) \ \boldsymbol{b}_1^\top \ \boldsymbol{b}_2^\top \ \ldots \ \boldsymbol{b}_L^\top \right]. \tag{3}$$

### 2.2 Attack model

We consider a white box attack, i.e., an adversary that has access to the architecture and the parameters $\boldsymbol{\theta}$ of the model. This could be the case, e.g., if the ML model is stored in an untrusted public cloud. We assume that the adversary cannot observe or interfere with the training process, and it cannot tamper with the process of labeling the training data, i.e., the attacker is honest-but-curious. In addition, to create shadow training datasets, it is assumed that the adversary has access to similar training data that can be used to train shadow classifiers that imitate the target classifier $\boldsymbol{f}_t$. The aforementioned assumption is common in methods involving a meta-classifier. In the absence of such data, synthetic or noisy real-world data could also be used to produce sufficiently accurate shadow classifiers such as in Shokri et al. [2017].

The objective of the attacker is to obtain an accurate estimate $\hat{\boldsymbol{p}}$ of the class label distribution of the data used to train the target classifier. We quantify the accuracy of the estimate using the KL divergence between the estimated and the true distribution,

$$D_{p_t}(\hat{\boldsymbol{p}}) = D_{KL}(\boldsymbol{p}_t || \hat{\boldsymbol{p}}) = \sum_{c=1}^C [\boldsymbol{p}_t]_c \log \left( [\boldsymbol{p}_t]_c / [\hat{\boldsymbol{p}}]_c \right), \tag{4}$$

which is a widely used measure for quantifying how much two distributions differ. The objective of the attacker is thus to minimize $D_{p_t}(\hat{p})$.

## 3 Accuracy-augmented permutation-invariant meta-classifier attack

The attack we propose in the following makes use of a meta-classifier that is trained to produce an estimate $\hat{p}$ of the training data class label distribution, when it is fed with the parameters $\theta$ of the target classifier $f_t$. Furthermore, the attack also utilizes an auxiliary data set denoted by $\mathcal{D}_{\text{aux}}$ containing $N_{\text{aux}}$ samples from each class $c$ to evaluate the classification accuracy of the target classifier model for data in each class. The accuracy on auxiliary data, denoted by $a \in [0,1]^C$, is also used as an input to the meta-classifier. We thus define the meta classifier as a function $g_{\text{MC}} : \mathbb{R}^{M+C} \to \Delta^{C-1}$, parametrized by a vector of weights $\omega_{\text{MC}}$, which are to be learned. The output of the meta-classifier is $\hat{p} = g_{\text{MC}}(\theta, a; \omega_{\text{MC}})$. Although the term 'classifier' is a misnomer, we retain the term due to the analogy to the meta-classifier method used in the literature.

### 3.1 Meta-classifier training data set

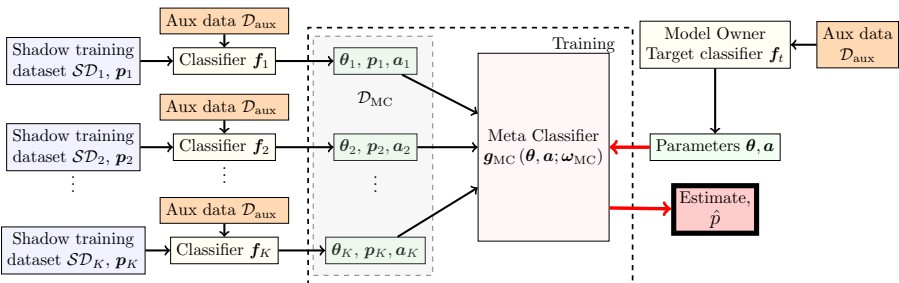

Figure 1: Shadow-training methodology for the accuracy augmented meta-classifier attack.

Figure 1 shows a flow chart of the proposed shadow-training methodology used for training the meta-classifier. First, a collection of shadow training datasets $\{\mathcal{SD}_k\}_k$, $k = 1, 2, \ldots, K$, is created; the class label distribution of data set $\mathcal{SD}_k$ is $p_k$, i.e., $\mathcal{SD}_k \triangleq \{x_i, e_i\}_{i=1,2,\ldots N}$ such that $\frac{1}{N} \sum_i e_i = p_k$. The number of shadow-training sets $K$ and the class label distributions $\{p_k\}_{k=1,2,\ldots,K}$ are parameters chosen by the attacker. The set of distributions $\{p_k\}$ can be chosen in various ways from $\Delta^{C-1}$, e.g., uniformly sampled or at random, as we discuss later in Section 3.3.

We use shadow-training dataset $\mathcal{SD}_k$ with class label distribution $p_k$ to train a shadow classifier $f_k$ with the same architecture as the target classifier $f_t$. The parameters $\theta_k$ of the shadow classifier $f_k$ trained using the shadow-training dataset $\mathcal{SD}_k$ are used as training data samples for the meta-classifier along with the class label distribution $p_k$ as the desired output. In addition, the accuracy of the shadow classifier $f_k$ on the auxilliary data, $a_k$, is also used as an input. The accuracy for class $c$ is defined as

$$[a_k]_c = (1/N_{\text{aux}}) \sum_{x_i \in \mathcal{D}_{\text{aux}} | [e_i]_c = 1} \mathbb{1}\{|[e_i - f_k(x_i; \theta_k)]_c| < \epsilon\}, \tag{5}$$

where $\mathbb{1}\{.\}$ is an indicator function which is 1 if the condition in its argument is true and zero otherwise, and $\epsilon$ is a threshold or meta-parameter used to assign the class label based on the output of the classifier. The idea behind using accuracy of the shadow classifier is that the accuracy on samples with a certain class-label is higher if that class is over-represented in the training dataset and therefore, additional information regarding accuracy could be beneficial in the inference of class-label distribution. The labeled training dataset for the meta-classifier is then defined as $\mathcal{D}_{\text{MC}} \triangleq \{\theta_k, p_k, a_k\}_{k=1,2,\ldots K}$.

Aligned with the goal of the attacker, discussed in Section 2.2, the meta-classifier is trained to minimize the KL-divergence between $p_k$ and the estimate $\hat{p}_k = g_{\text{MC}}(\theta_k, a_k; \omega_{\text{MC}})$ for all $K$ samples. Therefore, the loss-function of the meta-classifier is

$$\mathcal{L}(\omega_{\text{MC}}) = \sum_{k=1}^{K} D_{KL}(p_k || \hat{p}_k) = \sum_{k=1}^{K} \sum_{c=1}^{C} [p_k]_c \log([p_k]_c) - [p_k]_c \log([\hat{p}_k]_c). \tag{6}$$

Observe from (6) that the KL divergence consists of two terms, the cross entropy between distributions $\boldsymbol{p}_k$ and $\hat{\boldsymbol{p}}_k$, and the negative entropy of distribution $\boldsymbol{p}_k$. Of these two, only the cross-entropy term influences the update of weights $\boldsymbol{\omega}_{\mathrm{MC}}$, hence the loss function is effectively

$$\mathcal{L}\left(\boldsymbol{\omega}_{\mathrm{MC}}\right) = -\sum_{k=1}^{K}\sum_{c=1}^{C}[\boldsymbol{p}_k]_c \log\left([\boldsymbol{g}_{\mathrm{MC}}(\boldsymbol{\theta}_k, \boldsymbol{a}_k; \boldsymbol{\omega}_{\mathrm{MC}})]_c\right). \tag{7}$$

## 3.2 Accuracy-augmented permutation-invariant meta-classifier architecture

Recall that the training data set $\mathcal{D}_{\mathrm{MC}}$ for the meta-classifier consists of collections of weights and biases. It is thus important for the meta-classifier to be permutation invariant. Hence, we use DeepSets Zaheer et al. [2017] to obtain a succinct representation of the weights and biases $\boldsymbol{\theta}_k$ that are fed as input to the meta-classifier, similar to Ganju et al. [2018]. DeepSets allows the inputs to undergo a non-linear transformation before they are aggregated (summed) and transformed again. The transformation before aggregation allows one to create arbitrary moments of the inputs (e.g., second moment by squaring). The main advantage of this approach is that we learn a set-based representation that does not depend on the order of neurons in a certain layer. To obtain permutation invariance, we sum the transformed connection weights and biases of neurons in a layer thereby disregarding the ordering. The proposed meta-classifier architecture is organized as follows.

- A fully connected single layer network is employed to take as inputs the connection weights and biases of the $m_1$ neurons in layer $\ell = 1$. Let the weights of this single layer network be denoted by $\boldsymbol{\omega}_1 \in \mathbb{R}^{D+1}$. The output of the layer is

$$\boldsymbol{q}_1 = \boldsymbol{\phi}_1\left(\begin{bmatrix}\boldsymbol{W}_1^\top & \boldsymbol{b}_1\end{bmatrix}\boldsymbol{\omega}_1\right), \boldsymbol{q}_1 \in \mathbb{R}^{m_1} \tag{8}$$

where $\boldsymbol{\phi}_1$ is the activation function employed.

- For the subsequent layers $\ell = 2, 3, \ldots, L$ of weights and biases, along with the input of connection weights and biases of neurons, the output from the previous layer, $\boldsymbol{q}_{\ell-1} \in \mathbb{R}^{m_{\ell-1}}$ is used as an input. More specifically, let the combined input be

$$\boldsymbol{U}_\ell \triangleq \begin{bmatrix}\boldsymbol{W}_\ell \\ \boldsymbol{b}_\ell^\top\end{bmatrix} + \begin{bmatrix}\boldsymbol{q}_{\ell-1}\boldsymbol{1}^\top \\ \boldsymbol{0}\end{bmatrix} \tag{9}$$

where $\boldsymbol{1}$ is a vector of ones of size $m_\ell$. The addition of the outer product $(\boldsymbol{q}_{\ell-1}\boldsymbol{1}^\top)$ ensures that $\boldsymbol{q}_{\ell-1}$ is added to every column of the connection weight matrix $\boldsymbol{W}_\ell$. Elements in column $j$ of $\boldsymbol{W}_\ell$ corresponds to the connection weights from all neurons in layer $\ell - 1$ to neuron $j$ in layer $\ell$. Now, let the weights of the single layer network for layer $\ell$ be denoted by $\boldsymbol{\omega}_\ell \in \mathbb{R}^{m_{\ell-1}+1}$. Then its output is given by

$$\boldsymbol{q}_\ell = \boldsymbol{\phi}_\ell\left(\boldsymbol{U}_\ell^\top \boldsymbol{\omega}_\ell\right), \quad \boldsymbol{q}_\ell \in \mathbb{R}^{m_\ell} \tag{10}$$

- The transformed weights and biases from all layers are summed, i.e., $\sum[\boldsymbol{q}_1]_i, \sum[\boldsymbol{q}_2]_i, \ldots, \sum[\boldsymbol{q}_L]_i$, and the accuracy on the auxiliary dataset $\boldsymbol{a}$ are combined using another fully connected single layer network whose weights are denoted by $\boldsymbol{\Omega}_\rho \in \mathbb{R}^{L+C\times C}$. Finally, a softmax operation $\rho$ at the output of the fully connected layer ensures that the output $\hat{\boldsymbol{p}}$ of the meta-classifier is in the probability simplex $\boldsymbol{\Delta}^{C-1}$. The last layer of the meta-classifier is thus

$$\hat{\boldsymbol{p}} = \rho\left(\begin{bmatrix}\sum[\boldsymbol{q}_1]_i & \sum[\boldsymbol{q}_2]_i & \cdots & \sum[\boldsymbol{q}_L]_i & \boldsymbol{a}^\top\end{bmatrix}\boldsymbol{\Omega}_\rho\right). \tag{11}$$

To summarize, the parameters of the meta-classifier consist of the concatenation of $L$ vectors of weights, one for each layer and a final connection weight matrix at the output,

$$\boldsymbol{\omega}_{\mathrm{MC}} = \begin{bmatrix}\boldsymbol{\omega}_1^\top & \boldsymbol{\omega}_2^\top & \cdots & \boldsymbol{\omega}_L^\top & \mathrm{vec}(\boldsymbol{\Omega}_\rho)\end{bmatrix}. \tag{12}$$

Now, the proposed architecture for the meta-classifier is different from Ganju et al. [2018] in two crucial aspects. Firstly, the transformed weights and biases from the previous layer i.e. $\boldsymbol{q}_{\ell-1}$ are not concatenated to the weights and biases of layer $\ell$ as in Ganju et al. [2018]. Instead, in the proposed architecture, the combined input adds the output from the previous layer to every column of the connection weight matrix. This is done so that the the neuron $j$ in layer $\ell - 1$ and the transformed output $[\boldsymbol{q}_{\ell-1}]_j$ have the same parameter or weight $[\boldsymbol{\omega}_\ell]_j$ in the meta-classifier. Such an architecture,

reduces the number of parameters of the meta-classifier and also helps to combat over-fitting. In the proposed architecture, $\boldsymbol{\omega}_\ell$ is of size $m_\ell + 1$ whereas for architecture in Ganju et al. [2018], $\boldsymbol{\omega}_\ell$ would be of size $2m_\ell + 1$. Secondly, the accuracy on the auxiliary dataset is augmented in the last layer which is an addition to the architecture in Ganju et al. [2018]. We see that accuracy-augmentation leads to improvement over the architecture in Ganju et al. [2018].

Note that even though the architectures of the proposed meta-classifier and the one in Ganju et al. [2018] can be compared as discussed above, the attack goals accomplished by them are very different. In other words, the inference of class label distribution of training data is a novel attack.

### 3.3 Sampling schemes to select shadow training data sets

For a target binary classifier, the probability simplex $\boldsymbol{\Delta}^{C-1}$ is a line in $\mathbb{R}^2$ between points $(0,1)$ and $(1,0)$, and one needs to choose $\boldsymbol{p}_k$ from the line segment. The attacker could use uniform sampling on the line segment or it could sample at the ends of the line segment if it has prior information that the class label distribution is skewed, i.e., training data is imbalanced. However, sampling for multi-class classifiers implies choosing $\boldsymbol{p}_k$ from a higher dimensional probability simplex. For example, for a 3-class target classifier, a uniform sampling strategy amounts to choosing points uniformly on the grid which is an equilateral triangle formed in $\mathbb{R}^3$ with vertices $(0,0,1), (0,1,0), (1,0,0)$. Alternative strategies include sampling only on the edges of the triangle, and sampling close to the vertices. A major challenge in multi-dimensional sampling is that many samples are needed in order to have a certain minimum sample density for all the regions in the simplex, growing exponentially with the dimension. Since each sample $\boldsymbol{p}_k$ corresponds to a shadow training dataset, if the attacker has no prior information about the class label distribution used by the target classifier then the number of shadow datasets needed may render the approach infeasible. This is admittedly a limitation of the shadow-training approach for multi-class target classifiers. An empirical evaluation for 3-class classifier is presented in Appendix A.1 to understand what are sufficiently good sampling schemes to choose $\boldsymbol{p}_k$.

## 4 Numerical results

We present numerical results based on experiments conducted on two datasets, the UCI Census income dataset Dua and Graff [2017] and the MNIST handwritten digit classification dataset LeCun [1998]. For the UCI Census income dataset, the task is income classification as above or below $50$k USD which is a binary classification problem. For the MNIST dataset, we study both binary classification (between digits $0$ and $1$) and the multi-class classification. In the multi-class setting we consider classification among $3$ and $4$ digits (classes) because of the sampling complexity when generating the shadow-training datasets, as discussed in Section 3.3. We evaluate the benefit of accuracy augmentation and trimmed architecture by comparing the performance of the proposed meta-classifier trained with the accuracy on the auxiliary datasets with that of a meta-classifier whose architecture is akin to that in Ganju et al. [2018], albeit their goal of the meta-classifier is different from ours. We sample the probability simplex uniformly with step-size $\Delta p$ in each dimension to obtain the label proportions $\boldsymbol{p}_k$ for generating shadow-training datasets. The test set to evaluate the performance of the meta-classifier is made of class label distributions with the smallest step-size that was used to train the meta-classifier.

All the target classifiers $\boldsymbol{f}_t$ are fully-connected neural networks with $L = 3$ layers, ReLU activation function in the hidden layers, while the last layer has sigmoid and softmax for binary and multi-class classification, respectively. As a result, the accuracy-augmented meta-classifiers $\boldsymbol{g}_{\text{MC}}$ have $L = 3$ weight vectors and $1$ connection matrix overall as in (12) and the activation function $\phi_\ell$ is chosen as the ELU activation function (exponential linear unit) Clevert et al. [2015].

### 4.1 UCI Census income dataset

For the UCI Census income dataset, the target classifier is trained to classify the annual income of people being above or below 50k US dollars based on demographical and employment related feature inputs. Some of the feature inputs are categorical and therefore were converted into one-hot vectors. The 3 layers of the target classifier have $m_1 = 32$, $m_2 = 16$, $m_3 = 8$ neurons. Since the target is a

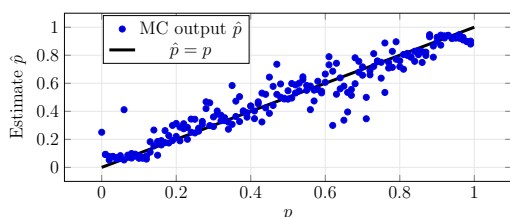
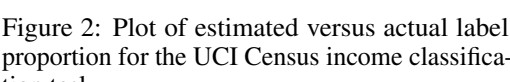
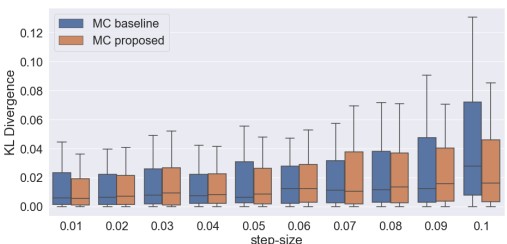

Figure 2: Plot of estimated versus actual label proportion for the UCI Census income classification task.

Figure 3: Performances of the proposed accuracy augmented meta-classifier and the meta-classifier with architecture as in Ganju et al. [2018] for step sizes $\Delta p$ for UCI Census income classification.

Table 1: Average MSE of meta-classifiers for UCI Census income classification for step sizes $\Delta p$.

| step-size | 0.01 | 0.02 | 0.03 | 0.04 | 0.05 | 0.06 | 0.07 | 0.08 | 0.09 | 0.1 |
|---|---|---|---|---|---|---|---|---|---|---|
| Baseline | 0.019 | 0.02 | 0.02 | 0.022 | 0.024 | 0.025 | 0.027 | 0.029 | 0.034 | 0.045 |
| Proposed | 0.016 | 0.017 | 0.02 | 0.02 | 0.02 | 0.024 | 0.024 | 0.026 | 0.031 | 0.032 |

binary classifier, we denote the fraction of the dataset used for training that has income greater than 50k USD by $p_k$ and the other class consequently has proportion $1 - p_k$.

The shadow classifiers were trained using shadow-training datasets with $N = 5000$ data points and with proportions $p_k$ uniform on $\mathbf{\Delta}^1$ with $\Delta p = 0.01$ i.e. $p_k = 0, 0.01, 0.02, \ldots 1$. Furthermore, for each value of $p_k$ we created 10 different datasets by resampling from the UCI Census income dataset to account for the heterogeneity of the feature inputs. Out of 10 data sets per proportion $p_k$, 8 were used for training and 2 for testing. Thus, overall we have $K = 800$ shadow training datasets and 200 test data sets. The proposed accuracy-augmented meta-classifier has 4 parameter vectors of dimensions as mentioned in Section 3.2. Figure 2 shows a scatter plot of the estimated label proportion $\hat{p}$ with respect to the actual label proportion $p$ for the UCI Census income dataset. The figure shows that the points are close to the $p = \hat{p}$ straight line, highlighting the efficacy of the proposed approach. Figure 3 shows the box-and-whisker plot of KL divergence between the estimated and actual label proportion distribution, $D_{p_t}(\hat{p})$, for the proposed accuracy-augmented meta-classifier and the baseline meta-classifier without accuracy-augmentation and architectural changes, as a function of the step size $\Delta p$. The test dataset contains 200 cases. The results show that the proposed accuracy-augmented meta-classifier outperforms the baseline meta-classifier, showing that the reduction in number of parameters and augmentation with accuracy of the auxiliary data is beneficial and improves the estimation capability of the meta-classifier. The improvement is especially prominent when $\Delta p$ is large. This implies that accuracy-augmentation and lower number of parameters allows the attacker to carry out attacks with a smaller shadow training dataset, i.e., with lower granularity, and still estimate the true label proportion. In order to gather more intuition, we also report the average mean-squared error in Table 1 between the estimated and true label distribution,

$$MSE = \|\boldsymbol{p}_k - \hat{\boldsymbol{p}}_k\|_2^2 \tag{13}$$

for the baseline meta-classifier and the proposed meta-classifier. A monotonic increase with respect to step-size is observed which is to be expected. The table shows a decrease in average MSE when using the proposed meta-classifier when compared to the baseline. In fact, the maximum decrease of MSE is 29.6% relative to the baseline for a step-size of 0.1.

## 4.2 MNIST dataset: Classification of digits 0 and 1

For binary classification on the MNIST dataset, we trained a binary target classifier for digits 0 and 1 in the dataset. The input pixels are arranged as a feature vector with dimension $D = 784$. The 3 layers for the target classifier have $m_1 = 128$, $m_2 = 32$, $m_3 = 16$ neurons. Since the target is a binary classifier, we denote the proportion of digit 0 used for training by $p_k$ and of digit 1 by $1 - p_k$. Similar to the UCI Census income dataset, the shadow-classifier datasets were also created with different

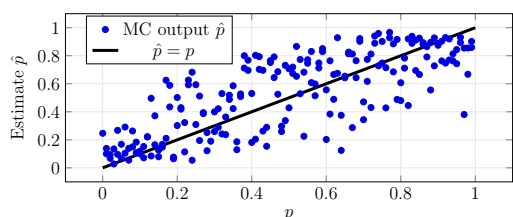

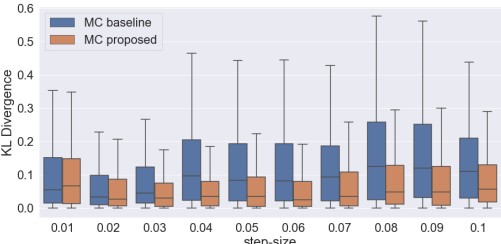

Figure 4: Plot of estimated versus actual label proportion for the MNIST Digit classification task between the digits 0 and 1.

Figure 5: Performances of the proposed accuracy augmented meta-classifier and the meta-classifier with architecture as in Ganju et al. [2018] for step sizes $\Delta p$ for MNIST digit classification $(0, 1)$

Table 2: Average MSE of meta-classifiers for MNIST digit classification $(0, 1)$ for step sizes $\Delta p$.

| step-size | 0.010 | 0.020 | 0.030 | 0.040 | 0.050 | 0.060 | 0.070 | 0.080 | 0.090 | 0.100 |
|---|---|---|---|---|---|---|---|---|---|---|
| Baseline | 0.080 | 0.068 | 0.090 | 0.123 | 0.131 | 0.127 | 0.126 | 0.164 | 0.175 | 0.172 |
| Proposed | 0.081 | 0.066 | 0.055 | 0.058 | 0.067 | 0.082 | 0.090 | 0.099 | 0.115 | 0.147 |

proportions $p_k = 0, 0.01, 0.02, \ldots 1$, with $N = 4000$ data points each. The rest of the evaluation methodology is the same as for the UCI Census income dataset binary classifier experiment. Figure 4 shows the estimated $\hat{p}$ versus actual label proportion $p$. Overall, the meta-classifier is not as accurate for MNIST as for the UCI Census income classifier. We attribute this to the higher complexity of the input feature space of MNIST, which makes the training of the meta-classifier less effective. However, the meta-classifier attack can still satisfactorily estimate the input label distribution for low and high $p$ values, i.e., when the training data set is unbalanced. Figure 5 shows the box-and-whisker plot of the KL divergence between the estimated and true label proportion distribution, $D_{p_t}(\hat{p})$, as a function of the sampling step-size $\Delta p$. As for the UCI Census dataset, the proposed accuracy-augmented meta classifier outperforms the baseline meta-classifier attack for all values of the step size $\Delta p$. The advantage of accuracy augmentation and reduced parameters is even clearer for MNIST as compared to UCI Census dataset. Table 2 also highlights the superiority of the proposed meta-classifier over the baseline since the average MSE is lower at all time steps. The maximum relative decrease with respect to the baseline is $52.6\%$ at step-size $0.04$.

## 5 Conclusions and future work

In this paper, we introduced a new type of property inference attack that aims at inferring the class label distribution of the training data from weights and biases of a trained classifier that has fully connected neural network architecture. We proposed a shadow-training based accuracy-augmented meta-classifier for this purpose. The empirical results on the UCI Income dataset and the MNIST dataset show the efficacy of the proposed method for class label distribution inference. Possible future research directions include exploring the potential of adaptive or online sampling schemes for shadow-training data sets improving the performance of multi-class meta-classifier attacks, and the development of mitigation schemes against such meta-classifier attacks. Future work also includes developing new meta-classifier architecture specific to other types of deep learning networks.

## Acknowledgments and Disclosure of Funding

This work was partly funded by the Vinnova Competence Center for Trustworthy Edge Computing Systems and Applications (TECoSA) at KTH and by the Swedish Foundation for Strategic Research through the CLAS project (grant RIT17-0046).

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

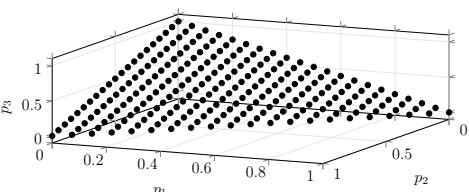

Figure 6: Samples of $p_k$ from the probability simplex $\mathbf{\Delta}^2$ for the MNIST 3-class classifier. $\Delta p = 0.05$

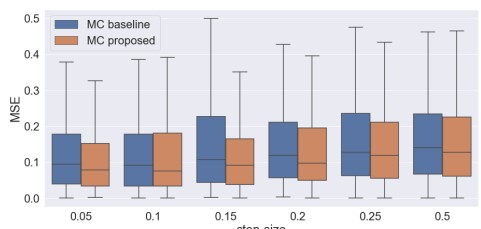

Figure 7: Attack performance of the proposed accuracy augmented meta-classifier and the meta-classifier with architecture as in Ganju et al. [2018] for 3-class MNIST digit classification (0, 1, 2) with uniform sampling from the probability simplex with sampling step size $\Delta p$.

Table 3: Table showing the average MSE of 3-class meta-classifier for MNIST Digit classification (0, 1, 2) 3-class with uniform sampling from the probability simplex and step size corresponds to sub-sampling from this simplex.

| step-size | 0.05 | 0.1 | 0.15 | 0.2 | 0.25 | 0.5 |
|---|---|---|---|---|---|---|
| Baseline | 0.126 | 0.131 | 0.145 | 0.153 | 0.17 | 0.165 |
| Proposed | 0.11 | 0.116 | 0.123 | 0.137 | 0.155 | 0.163 |

## A Appendix

### A.1 Multi-class target classifier

For the multi-class target classifier, we use the MNIST dataset with 3 and 4 digits for classification. For the 3-class and 4-class classifier we used digits $0, 1, 2$ and $0, 1, 2, 3$, respectively. The target classifier for the multi-class MNIST classifier has the same architecture as the binary MNIST classifier with the only difference being the number of outputs and the use of softmax as the activation function at the last layer. The architecture of the meta-classifier attack is also the same as in the binary MNIST case, with the difference that the number of outputs is $C$ for ($C = 3$ and $C = 4$, respectively), and softmax is the activation function $\rho$ at the last layer of the meta-classifier. Figure 6 shows the sampled points $p_k$ for the uniformly sampled label distributions employed to generate data for the shadow classifiers with different $p_k$. As discussed in Section 3.3, for the 3-class target classifier, $p_k$ must be drawn from the probability simplex $\mathbf{\Delta}^2$. Figure 7 shows the box-and-whisker plot of KL divergence between the actual and the estimated distributions for the meta-classifier with accuracy augmentation and without, as a function of the step size $\Delta p$. While the overall accuracy of the attack is worse than that for binary classification, the proposed accuracy augmentation is clearly beneficial for the attack. Table 3 shows the average MSE for the proposed and the baseline meta-classifier, and shows an improvement at all step sizes; the maximum decrease relative to the baseline is $14.9\%$ at step-size $0.15$.

To explore the influence of the sampling pattern of $p_k$ on shadow training, we investigate two different sampling patterns: a) $p_k$ sampled in the middle of the probability simplex and b) $p_k$ sampled along the 3 edges of the probability simplex. In Figure 8 and Figure 9 we show a heatmap of the KL divergence of the estimated distribution with respect to the actual distribution, $D_p = D_{KL}(p||\hat{p})$ with respect to the true distribution $p$. For easier visualization, only the first 2 dimensions of $p$ are plotted since the third dimension is dependent on the first two, $[p]_3 = 1 - [p]_1 - [p]_2$. The color is proportional to the KL divergence $D_p$. Figure 8 shows that when the meta-classifier is trained on samples from the middle of the probability simplex, the KL divergence is lower in the middle than along the edges, i.e. when values of $[p]_1$ and $[p]_2$ are either large or small. However, in Figure 9, even though edges of the probability simplex were sampled to train the meta-classifier, the KL divergence is still high at the edges. The overall range of KL divergences seems to be smaller in this case compared to the one sampled in the middle. Overall, it seems that sub-sampling from the probability simplex for the sake

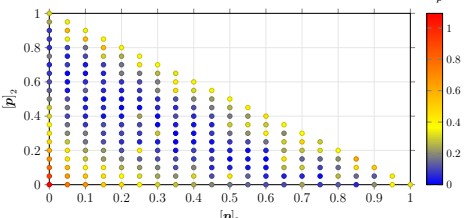
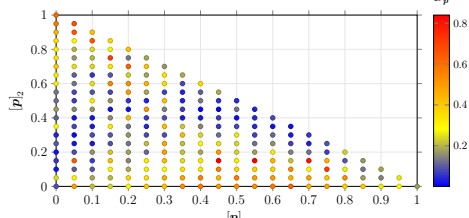

Figure 8: KL Divergence $D_{KL}(\boldsymbol{p}||\hat{\boldsymbol{p}})$ with respect to the first 2 dimensions of true distribution $\boldsymbol{p}$ when the meta-classifier was trained on samples from the middle of the simplex. $\Delta p = 0.05$

Figure 9: KL Divergence $D_{KL}(\boldsymbol{p}||\hat{\boldsymbol{p}})$ with respect to the first 2 dimensions of true distribution $\boldsymbol{p}$ when the meta-classifier was trained on samples at the 3 edges of the probability simplex. $\Delta p = 0.05$

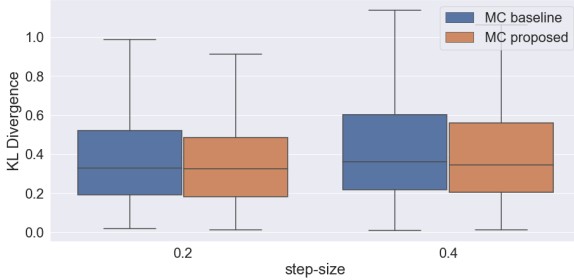

Figure 10: Attack performance of the proposed accuracy augmented meta-classifier and the meta-classifier with architecture as in Ganju et al. [2018] for 4-class MNIST digit classification (0, 1, 2, 3) with uniform sampling from the probability simplex with sampling step size $\Delta p$.

of training does not adversely affect the ability of the attacker to infer label distributions. Therefore, an attacker could potentially choose a region from the probability simplex based on prior knowledge and only train the meta-classifier using shadow classifiers with label distribution sampled from this region. This is one of the ways to reduce the sampling complexity.

For the 4-class classifier between digits $0, 1, 2, 3$, we trained the meta-classifier on samples $\boldsymbol{p}_k$ uniformly sampled from the $\boldsymbol{\Delta}^3$ simplex with step sizes $\Delta p = 0.2, 0.4$. Figure 10 shows the box-and-whisker plots of the KL divergences obtained for the baseline and the proposed meta-classifiers. The overall range of values is smaller when compared to the baseline. For better understanding, we also report the average MSE for the uniform sampling case in Table 4 for the proposed meta-classifier and the baseline. From these values, it is evident that accuracy augmentation and architectural changes are beneficial even in higher dimensional settings, and the maximum decrease in MSE relative to the baseline is $5.4\%$ at step-size $0.4$. However, we see that the overall performance has slightly deteriorated in comparison to binary and 3-class classifiers. This is to be expected given the dimensionality of the problem and the computational difficulty of producing shadow training datasets that are representative of all possible label proportions, due to the large number of samples required for the same sampling density as the 3-class classifier. A potential approach to address the issue of sampling complexity would be to use prior knowledge and only sample from certain regions from the probability simplex in higher dimensions which is the subject of our future work.

## A.2 Random oversampling as a countermeasure

The imbalance of class labels is often addressed prior to training a classifier by random oversampling of the minority class Chawla et al. [2002], Ling and Li [1998]. Through random oversampling, the label distribution used for training becomes uniform, and thus the true label distribution $\boldsymbol{p}$ becomes hidden. Hence, random oversampling could be a simple countermeasure to the considered class label distribution inference attacks. However, since random oversampling leads to the repetition of data samples from minority classes, the parameters of the target classifier could still be similar to the parameters of a target classifier trained on the original imbalanced dataset.

Table 4: Average MSE divergence of 4-class meta-classifiers for MNIST Digit classification (0, 1, 2, 3) with uniform sampling from the probability simplex and step sizes $\Delta p$.

| step-size | Baseline | Proposed |
|-----------|----------|----------|
| 0.200 | 0.174 | 0.173 |
| 0.400 | 0.201 | 0.190 |

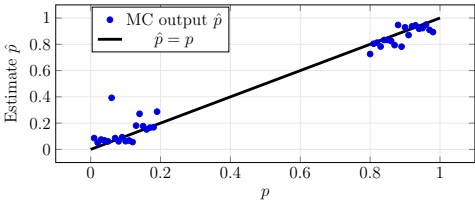

Figure 11: Estimates of label distribution $\hat{p}$ for imbalanced data sets adjusted by random oversampling versus original label distribution.

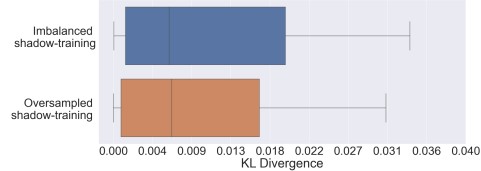

Figure 12: Performance of the proposed meta-classifier with (oversampled shadow-training) and without training (imbalanced shadow-training) for target classifiers that were trained on data sets adjusted for class imbalance using random minority class oversampling.

We now investigate if class label distribution attacks are effective despite random oversampling, i.e., if the true label distribution $p$ can be estimated by a meta-classifier based on the parameters of a target classifier that was instead trained on over-sampled data with induced uniform label distribution $p_o$. We use the UCI Census income dataset and binary classification for the evaluation. The data sets with label distributions $p \in (0, 0.2] \cup [0.8, 1)$ were considered imbalanced and used those for training target classifiers by oversampling the minority class, i.e., $p_o = 0.5$. We then used the accuracy-augmented meta-classifier that was trained in Section 4.1 with $\Delta p = 0.01$ in order to infer the class label distribution $p$, i.e., the attacker is not aware of random oversampling. Figure 11 shows the estimated label distribution plotted with respect to the original label distribution $p$. It shows that the proposed meta-classifier can successfully infer the true label distribution despite oversampling, hence oversampling may not be an effective countermeasure against the proposed meta-classifier attack. In addition, if the attacker is aware of random oversampling, it could re-train the meta-classifier by generating shadow-training data sets with random oversampling and labels as the original imbalanced distribution and further improve the inference accuracy. This is verified next. We consider training the meta-classifier to be aware of random oversampling, i.e., in the shadow training data set for the meta-classifier, the parameters of the shadow classifiers that are trained on random oversampling adjusted shadow training data sets are labeled with the original label distribution. As before, this experiment is performed on the UCI Census income data. Figure. 12 shows the KL divergence for a meta-classifier that is not aware of random oversampling and for one that is aware of it. The figure shows that the meta classifier that was trained on the oversampled shadow training data sets achieves better accuracy, i.e. lower KL divergence.

### A.3 Limitations of the Attack

The limitations of the proposed meta-classifier attack can be grouped into two categories: the architecture of the target classifier and the shadow-training methodology. The proposed methodology is applicable to deep architectures with many layers. However, if the target classifier is a convolutional neural network (CNN), permutation invariance does not hold so one would need appropriate meta-classifier architectures other than DeepSets to also encode the spatial dependencies of the filter weights and biases. In Parisot et al. [2021], a multi-layer perceptron is proposed as meta-classifier when the target classifier is a CNN, but it unfortunately does not perform the encoding and could lead to loss of attack efficacy. A direction of future work would be to develop specific meta-classifiers for CNNs that perform well. The shadow training methodology itself has limitations, as it does not scale well with the number of classes (as discussed in Section 3.3) and with the number of model parameters in the target classifier. Increasing the number of classes requires more shadow training data sets, hence longer training time, while increasing the number of model parameters increases the number of parameters of the meta classifier, making it more challenging to be trained.

