# OpenReview forum: "Inferring Class Label Distribution of Training Data from Classifiers: An Accuracy-Augmented Meta-Classifier Attack"
_NeurIPS.cc/2022/Workshop/TSRML — TSRML2022_

### Official Review · Reviewer_CLJH · 2022-10-12
**This paper introduces a new property inference attack that infers the class label distribution of the training data used by the target classifier.**

**Overall Rating:** 6

**Summary:**

This paper designs a strong privacy attack against trained classifiers by estimating the class label distribution of the training set. To this end, the authors use shadow training and a meta-classifier in the white-box setting. This paper also leverages the accuracy of the classifier to augment the estimation performance. The authors use a DeepSet to design a permutation-invariant architecture, which reduces the parameter number.

**Strengths:**

This paper has the following strengths:
+ The proposed property inference attack is stronger than the previous binary classification-like attack since it recovers the class label distribution of the training data.
+ The authors combine two features of the classifiers for the inference attack, i.e., the model parameters and the accuracy.
+ The authors design a permutation-invariant architecture for the meta-classifier using DeepSet and modify the forward pass flow to reduce the parameter count.

**Weaknesses:**

This paper has the following weakness points:
- The authors shall discuss the connection/difference between the proposed attack and membership inference attacks. Prior works have shown membership inference attacks using the model weights or the model's prediction entropy/loss as the features. Assuming the membership inference attack is successful, the adversary can combine the recovered membership information and the model's prediction to estimate the class label distribution as well (which is the focus of this work).
- The experiments performed in the paper are relatively small-scale. Particularly, the target classifier is a simple model with three fully-connected layers. It is not clear if the proposed attack will be successful against complex models with many layers. Also, the overhead and scalability of the attack are not discussed in the paper.

**Overall Recommendation:**

Overall, this paper presents a new and effective inference attack against pre-trained models using shadow training and meta-classifier. The authors have proposed several optimization techniques to improve the attack performance.

**Review Confidence:**

3: The reviewer is fairly confident that the evaluation is correct

---

### Official Review · Reviewer_nFgB · 2022-10-19

**Overall Recommendation:** I believe this is a meaningful study …
**Overall Rating:** 6

**Summary:**

In the paper, the authors studied the problem of property inference attacks. In this setting, an adversary wants to infer the class label distribution of training data purely from a trained classifier.
Usually, this procedure is achieved by a meta-classifier that maps the target model's parameters to certain sensitive attributes, such as the training class distribution. Specifically, the attacker has the access to a dataset that is similar to the original training set. To create a permutation invariant meta-classifier, the authors followed the DeepSets to translate and concatenate the weights.
The authors demonstrated the empirical evaluation on a few datasets. The proposed method can effectively  infer the training data distribution and outperforms a few baselines.

**Strengths:**

-The paper is clearly written and easy to follow.
-The proposed study, including the property inference attack, as far as I can tell, is quite novel.

**Weaknesses:**

Typically, people will do data augmentation and/or regularization to promote the performance of classification models. I am wondering how such factors are considered.

For more complicated datasets, people will use more sophisticated deep neural networks, including convolution neural networks and transformers. Thus it will become harder to formulate a meta-learner in this case.

I believe this problem looks like an inverse problem of neural architecture search or AutoML. Is it possible to borrow some techniques, such as BO, from these studies?

**Review Confidence:**

4: The reviewer is confident but not absolutely certain that the evaluation is correct

---

### Decision · Program_Chairs · 2022-10-23

Accept